

# Study of miRNA and lymphocyte subsets as potential biomarkers for the diagnosis and prognosis of gastric cancer

Jinpeng Li[*], Zixi Chen[*], Qian Li, Rongrong Liu, Jin Zheng, Qing Gu, Fenfen Xiang, Xiaoxiao Li, Mengzhe Zhang, Xiangdong Kang and Rong Wu

Department of Laboratory Medicine, Putuo Hospital, Shanghai University of Traditional Chinese Medicine, Shanghai, China

[*] These authors contributed equally to this work.

Corresponding authors
Xiangdong Kang, xd_kang@163.com
Rong Wu, rong701@126.com

## ABSTRACT

**Objective**. The aim of this study was to identify the expression of miRNA and lymphocyte subsets in the blood of gastric cancer (GC) patients, elucidate their clinical significance in GC, and establish novel biomarkers for the early diagnosis and prognosis of GC.

**Methods**. The expression of miRNAs in the serum of GC patients was screened using second-generation sequencing and detected using qRT-PCR. The correlation between miRNA expression and clinicopathological characteristics of GC patients was analyzed, and molecular markers for predicting cancer were identified. Additionally, flow cytometry was used to detect the proportion of lymphocyte subsets in GC patients compared to healthy individuals. The correlations between differential lymphocyte subsets, clinicopathological features of GC patients, and their prognosis were analyzed statistically.

**Results**. The study revealed that hsa-miR-1306-5p, hsa-miR-3173-5p, and hsa-miR-296-5p were expressed at lower levels in the blood of GC patients, which is consistent with miRNA-seq findings. The AUC values of hsa-miR-1306-5p, hsa-miR-3173-5p, and hsa-miR-296-5p were found to be effective predictors of GC occurrence. Additionally, hsa-miR-296-5p was found to be negatively correlated with CA724. Furthermore, hsa-miR-1306-5p, hsa-miR-3173-5p, and hsa-miR-296-5p were found to be associated with the stage of the disease and were closely linked to the clinical pathology of GC. The lower the levels of these miRNAs, the greater the clinical stage of the tumor and the worse the prognosis of gastric cancer patients. Finally, the study found that patients with GC had lower absolute numbers of CD3+ T cells, CD4+ T cells, CD8+ T cells, CD19+ B cells, and lymphocytes compared to healthy individuals. The quantity of CD4+ T lymphocytes and the level of the tumor marker CEA were shown to be negatively correlated. The ROC curve and multivariate logistic regression analysis demonstrated that lymphocyte subsets can effectively predict gastric carcinogenesis and prognosis.

**Conclusion**. These miRNAs such as hsa-miR-1306-5p, hsa-miR-3173-5p, hsa-miR-296-5p and lymphocyte subsets such as the absolute numbers of CD3+ T cells, CD4+ T cells, CD8+ T cells, CD19+ B cells, lymphocytes are down-regulated in GC and are closely related to the clinicopathological characteristics and prognosis of GC patients. They may serve as new molecular markers for predicting the early diagnosis and prognosis of GC patients.

## INTRODUCTION

Gastric cancer (GC) is a prevalent malignant tumor affecting the digestive tract, ranking sixth globally in terms of new tumor cases and second in terms of mortality rates (*Li, Doherty & Wang, 2022*; *Sexton et al., 2020*; *Wei et al., 2020*). China, in particular, experiences a high incidence of GC, accounting for 42.6% of global cases and 45.0% of global deaths related to GC (*Feng et al., 2020*; *Song et al., 2011*). According to the most recent tumor statistics released by the National Cancer Center in 2018, China has the third-highest incidence rate (41/10,000) and the second-highest mortality rate (29.4/10,000) for GC (*Zhao et al., 2021*). These data suggest that GC poses significant challenges in terms of treatment compared to other tumor types (*Zhao et al., 2021*).

Due to the absence of early noticeable symptoms, most GC patients are diagnosed at advanced stages when treatment options become limited. Although surgical, chemotherapeutic, and targeted treatment strategies for GC have improved, the overall survival rate remains unfavorable. Currently, gastroscopy is considered the gold standard for diagnosing GC (*Chemaly et al., 2022*; *Lee et al., 2022*). However, its cost and invasiveness limit its widespread use. Therefore, there is an urgent need for non-invasive or minimally invasive serum markers that can be readily employed in clinical practice.

Currently, traditional serum markers associated with GC primarily include carcinoembryonic antigen (CEA), carbohydrate antigen CA199, and carbohydrate antigen CA724. While these markers provide some predictive value regarding the occurrence, recurrence, and metastasis of GC, their specificity and sensitivity are insufficient for diagnosing early-stage GC and effectively predicting recurrence and metastasis (*Chen et al., 2022*; *So et al., 2021*; *Zhao et al., 2022*). Consequently, it is crucial to further investigate molecular markers of potential clinical significance to enhance the early detection rate and prognosis of GC patients.

MicroRNAs (miRNAs) are a class of endogenous non-coding small RNAs consisting of approximately 22 nucleotides. They primarily regulate gene expression by binding to target gene mRNA, leading to mRNA degradation or translational inhibition (*Kilikevicius, Meister & Corey, 2022*). Numerous studies have demonstrated that miRNAs can be stably detected in various tissues, blood, stool, saliva, and ascites, making them easily accessible for testing (*Bendifallah et al., 2022*; *Reithmair et al., 2017*; *Yang et al., 2022*). Emerging evidence suggests that miRNAs can function as tumor suppressors or oncogenes, with changes in miRNA expression playing a pivotal role in tumorigenesis and cancer progression, particularly in GC (*Guo et al., 2022a*; *Liu et al., 2022*; *Smolarz et al., 2022*). It has been shown by *Li et al. (2010)* that a few miRNA signature (miR-10b, miR-21, miR-126, miR-30a-5p, miR-338, let-7a, and miR-223) is an independent predictor of overall survival and relapse-free survival in GC. A short survival team is also characterized by high expression of miR-150, miR-20b, miR-142-5p, miR-214, and miR-375 and low

expression of miR-433, miR-451, let-7g and miR-125-5p (*Nishida et al., 2011*; *Wang et al., 2013*). There is a strong association between onco-miR-10b, miR-21, and miR-212 expression in gastric cancer patients and metastasis as well as poor clinical outcomes (*Xiong et al., 2011*). *Zhang et al.*'s (*2020*) research revealed three potential biomarkers, namely miR-10b-5p, miR-101-3p, and miR-143-5p, for different types of gastric cancer metastasis. Specifically, miR-10b-5p was proposed as a biomarker for gastric cancer with lymph node metastasis, miR-101-3p for gastric cancer with ovarian metastasis, and miR-143-5p for gastric cancer with liver metastasis. Plasma or serum miRNAs, which exhibit remarkable stability, have been identified as promising candidates for biomarker screening. The stability of circulating miRNAs in serum and their aberrant expression in GC have been well-established, positioning miRNAs as potential diagnostic and prognostic biomarkers for various types of cancer, including GC. Numerous studies have revealed a close association between the occurrence and progression of tumors and the immune status of the host (*Lei et al., 2022*; *Xing et al., 2022*; *Zhang et al., 2022*). Cellular immunity, especially T lymphocyte subsets, plays a critical role in the body's antitumor immune response (*Peng et al., 2022*; *Qi et al., 2022*). Among the T cell subsets, CD3+ represents T lymphocytes, which are further divided into CD4+ helper/inducible T cells and CD8+ suppressive/cytotoxic T cells. CD4+ cells assist B cells in antibody secretion and regulate the immune response of other T cells, while CD8+ cells exhibit cytotoxic activity and play a crucial role in eliminating virus-infected and tumor cells (*Qi et al., 2022*). In recent years, it has been reported that the absolute peripheral blood counts of CD3+, CD4+, and CD4+/CD8+ cells were significantly lower in tumor patients compared to healthy individuals (*Huang et al., 2022*; *Notarangelo et al., 2022*). The alteration in the absolute counts of T lymphocyte subsets is believed to hold significant importance in the monitoring and prognosis of cancer patients, warranting further investigation.

In this study, our objective was to identify the expression of miRNAs and their diagnostic value in GC. We analyzed the miRNA expression profile using second-generation sequencing and bioinformatics methods, aiming to predict their potential molecular mechanisms. Furthermore, we compared the expression of miRNAs in the blood of GC patients with that of healthy individuals and investigated their relationship with clinicopathological characteristics. Additionally, we examined the proportion of lymphocyte subsets in the blood of GC patients compared to a healthy control group. Finally, we employed Receiver Operating Characteristic (ROC) curves to assess the potential of these miRNAs and lymphocyte subsets as novel biomarkers for the diagnosis and prognosis of GC patients.

## MATERIALS AND METHODS

### Patient inclusion criteria and blood sample collection

A prospective study was conducted involving a total of 20 GC patients with gastric adenocarcinoma and 20 healthy adult volunteers. Thirteen of these patients with stomach cancer had tumors that were clinically diagnosed at stage III–IV, and seven of them had stage I–II tumors. The patients were admitted to the General Surgery Department of Pu

Tuo Hospital, which is affiliated with Shanghai University of Traditional Chinese Medicine in Shanghai, China. The diagnosis of GC was confirmed by more than two professional pathologists following the patients' initial radical operation. We established inclusion criteria for patients with gastric cancer: (1) Prior to the operation, all patients did not undergo anti-tumor treatments such as radiotherapy, chemotherapy, and immunotherapy. Complete clinical and pathological data, as well as follow-up information, were available for all patients. (2) Pregnant women and children were excluded from the study, and patients with extensively metastatic tumors or other tumors were also excluded. (3) All selected patients provided informed consent for participation in the research, and the study adhered to the ethical principles outlined in the World Health Organization Helsinki Declaration. This study was approved by the Ethics Committee of Pu Tuo Hospital Affiliated to Shanghai University of Traditional Chinese Medicine, Shanghai, China. Informed consent was obtained from the patients (No. PTEC-A-2022-31-1) and written informed consent was received.

Serum blood samples were collected from the 20 GC patients who were admitted to Shanghai University of Traditional Chinese Medicine between January 2021 and January 2022. Among them, there were 13 males and seven females, with an average age of ($70.55 \pm 1.66$) years. Additionally, 20 serum samples were collected from healthy individuals during the same period, consisting of 16 males and four females, with an average age of ($66.48 \pm 1.97$) years. The distribution of age and gender between the two groups did not differ statistically significantly ($P > 0.05$). The patients' pathological types and characteristics were recorded. Each blood sample was promptly centrifuged, and the serum was stored at $-80\,°C$ until further analysis.

## Instruments and reagents

The miRNA extraction kit was obtained from Beijing Tiangen Biotech Co., Ltd. The primers for hsa-miR-1306-5p, hsa-miR-1307-5p, hsa-miR-3173-5p, hsa-miR-424-5p, hsa-miR-651-5p, and hsa-miR-296-5p were synthesized by Sangon Biotech Co., Ltd. (Shanghai, China). The fluorescence quantitative PCR instrument used was the Applied Biosystems 7300, obtained from Applied Biosystems Co., Ltd. (Waltham, MA, USA). The lymphocyte subpopulation detection six-color antibodies and BD Trucount absolute counter tubes were purchased from BD Co (East Rutherford, NJ, USA). The flow cytometry equipment used was the BD FACSCanto, also obtained from BD Co. (East Rutherford, NJ, USA).

## Validation of miRNAs by qRT-PCR

Serum samples were collected from the patients and divided into 1.5 mL Eppendorf tubes. They were then stored in a refrigerator at $-80\,°C$ for future use. MiRNAs was extracted from the serum using the miRNA extraction kit, and the concentration and purity of the extracted RNA were determined by Thermo Scientific NanoDrop (Thermo Fisher Scientific, Waltham, MA, USA). The quality and concentration of the miRNAs extracted in this study met the requirements for PCR reactions. Using U6 as the internal reference gene, the levels of hsa-miR-1306-5p, hsa-miR-1307-5p, hsa-miR-3173-5p, hsa-miR-424-5p,

**Table 1    Primer sequences of six miRNAs and U6.**

| Target | Primer sequences |
|---|---|
| has-miR-296-5p | ATTAAGGGCCCCCCCTCAAT |
| has-miR-1306-5p | CCACCTCCCCTGCAAACGTCCA |
| has-miR-3173-5p | CGCTGCCCTGCCTGTTTTCTCC |
| hsa-mir-424-5p | GCCCGCCAGCAGCAATTCATGT |
| hsa-mir-651-5p | CGCCCGCGTTTAGGATAAGCTTGACT |
| has-miR-1307-5p | ATAATCTCGACCGGACCTCGA |
| U6 | CTCGCTTCGGCAGCACA |
| | AACGCTTCACGAATTTGCGT |

hsa-miR-651-5p, and hsa-miR-296-5p in the serum were detected using qRT-PCR. The reaction procedure of qRT-PCR was set as described in previous article.The relative expression levels were calculated using the $2^{-\Delta\Delta Ct}$ method. The primer sequences for all miRNAs and U6 are listed in Table 1.

## MiRNA sequencing analysis

The total RNA was extracted from the patients' serum using the Trizol reagent, following the instructions provided by the manufacturer (Life Technologies, Carlsbad, CA). To assess RNA integrity and DNA contamination, denatured agarose gel electrophoresis was employed. Prior to constructing the RNA sequencing (RNA-seq) library, the Ribo Zero rRNA removal kit (Illumina, San Diego, CA, USA) was utilized to eliminate rRNA, while the miRNA enrichment kit (Cloud seq, USA) was employed to enrich miRNA. The RNA-seq library was constructed according to the manufacturer's instructions, using the TruSeq chain total RNA library preparation kits (Illumina, San Diego, CA, USA). Subsequently, all small RNAs were sequenced, potential miRNAs were extracted through data processing, and their expression levels were analyzed. The statistical power of this experimental design, calculated in RNASeqPower (An online implementation of RNASeqPower is available at https://rodrigo-arcoverde.shinyapps.io/rnaseq_power_calc/) is 0.81. There are 10 biological and technical replicates used to achieve the claimed statistical power.

## Detection of CEA, CA199 and CA724

Three tumor markers were measured using a chemiluminescence immunoassay. The equipment used was the Roche i2000 Luminous Immunometer (Basel, Switzerland). The corresponding reagents provided by Roche were used. The procedures were strictly performed following the instructions provided with the kit.

## Flow cytometry analysis

Peripheral blood samples (two mL) were collected from 51 GC patients and 50 healthy individuals in EDTA anticoagulant tubes. The lymphocyte subsets, including CD3+, CD4+, CD8+ T cells, CD19+ B cells, and CD16+ CD56+ NK cells, were quantified in cells/μL using a six-color flow cytometry assay. Human monoclonal antibodies, including anti-CD3-fluorescein isothiocyanate (FITC), anti-CD4-phycoerythrin (PE), anti-CD8-allophycocyanin (APC), anti-CD19-PE, anti-CD16-APC, and anti-CD56-PE

(BD Multitest), were used according to the manufacturer's instructions. The cell samples were analyzed on a BD Canto II flow cytometry system (BD Biosciences, East Rutherford, NJ, USA).

## Statistical analysis

All statistical analyses were conducted using SPSS 21.0 (Chicago, IL, USA). Data conforming to normal distribution are expressed as mean ± standard error of the mean (SEM) and were compared using the independent samples $t$-test. Data not conforming to normal distribution were expressed as median ± interquartile range (IQR) and compared using the Mann–Whitney test for nonparametric tests. Categorical variables are described as frequencies and percentages for each category and were analyzed using the Chi-square test or Fisher's exact test. A two-sided $p$-value of $<0.05$ was considered statistically significant. Binary logistic regression analysis was employed to identify independent risk factors associated with severity. Pearson's correlation analysis was used to examine the relationships between miRNAs, peripheral lymphocyte subsets and CA724, CA199, and CEA. The diagnostic value of the selected parameter in differentiating between mild and severe cases was evaluated using logistic regression analysis, receiver operating characteristic (ROC) curve, and the area under the ROC curve (AUC). Cutoff values were determined based on Youden's index from the ROC curve.

## RESULTS

### Results of miRNA sequencing in GC and healthy groups

In this experiment, high-throughput miRNA sequencing was performed to screen for differential miRNA in the serum of five GC patients and five healthy individuals. The screening data were first standardized, and the miRNA expression was visualized using a box plot. The plot displays the minimum value, the first quartile (25%), the median (50%), the third quartile (75%) and the maximum value, indicating relatively symmetrical data distribution with less scatter (Fig. 1A). Differential analysis of genes was conducted using a differential analysis tool, calculating the $P$-value and fold change (FC) for each gene. Based on the criteria of FC $\geq 2$ (up-regulated) or $\leq 0.5$ (down-regulated) and $p < 0.05$, a total of 50 differentially expressed miRNAs were identified, including 33 up-regulated miRNAs and 17 down-regulated miRNAs (Fig. 1B). Subsequently, gene ontology (GO) and Kyoto Encyclopedia of Genes and Genomes (KEGG) analyses revealed that the PI3K-Akt signaling pathway, Human papillomavirus infection, MAPK signaling pathway, and Proteoglycans in cancer pathway were relatively enriched. These pathways are known to play crucial roles in GC development (Fig. 1C). From these pathways, six key miRNAs, namely hsa-miR-1306-5p, hsa-miR-1307-5p, hsa-miR-3173-5p, hsa-miR-424-5p, hsa-miR-651-5p, and hsa-miR-296-5p, were selected for further validation as they were simultaneously associated with the four pathways relevant to GC. Among these miRNAs, hsa-miR-1306-5p, hsa-miR-3173-5p, and hsa-miR-296-5p showed significantly higher expression in the healthy group compared to the GC group. Conversely, the signal values of hsa-miR-424-5p, hsa-miR-651-5p, and hsa-miR-1307-5p were significantly higher in the GC group compared to the healthy group (Fig. 1D).
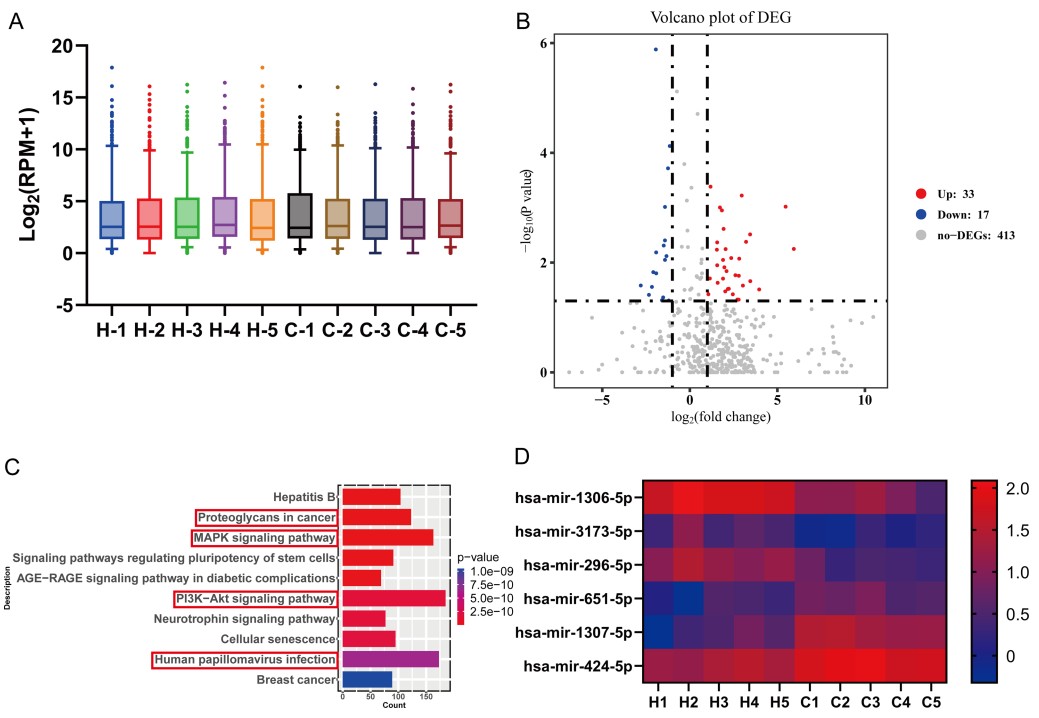

**Figure 1** **High-throughput miRNA sequencing was performed to screen for differential miRNA in the serum of five GC patients and five healthy individuals.** (A) The screening data were first standardized, and the miRNA expression was visualized using a box plot. The plot displays the minimum value, the first quartile (25%), the median (50%), the third quartile (75%) and the maximum value, indicating relatively symmetrical data distribution with less scatter. (B) Differential analysis of genes was conducted using a differential analysis tool, calculating the *P*-value and fold change (FC) for each gene. (C) Gene ontology (GO) and Kyoto Encyclopedia of Genes and Genomes (KEGG) analyses were taken to predict the target genes for differential miRNAs. (D) Six key miRNAs with high reliability related to gastric cancer were screened through bioinformatics analysis, with red indicating high expression genes and blue indicating low expression genes.

## Validation of miRNA sequencing results by qRT-PCR

We performed miRNA detection using a miRNA chip on blood samples from 20 GC patients and 20 healthy individuals. According to the qRT-PCR results, the expression levels of the hsa-miR-1306-5p, hsa-miR-3173-5p, hsa-miR-296-5p, hsa-miR-424-5p, hsa-miR-651-5p, and hsa-miR-1307-5p were considerably greater in the healthy group than the GC group (Figs. 2A–2F, expressed as median ± IQR). The qRT-PCR results were consistent with the miRNA-seq data for hsa-miR-1306-5p, hsa-miR-3173-5p, and hsa-miR-296-5p. Further analysis of the qRT-PCR data in conjunction with clinicopathological characteristics of GC patients revealed that the expression levels of hsa-miR-1306-5p, hsa-miR-3173-5p, and hsa-miR-296-5p were associated with the stage of GC. Through comprehensive analysis of qRT-PCR data and the clinical pathological characteristics of gastric cancer patients, we have identified a correlation between the expression of hsa-miR-1306-5p, hsa-miR-3173-5p, and hsa-miR-296-5p and the staging of patients. As the tumor stage advances, the expression levels of these miRNAs decrease significantly (Fig. 3A, expressed

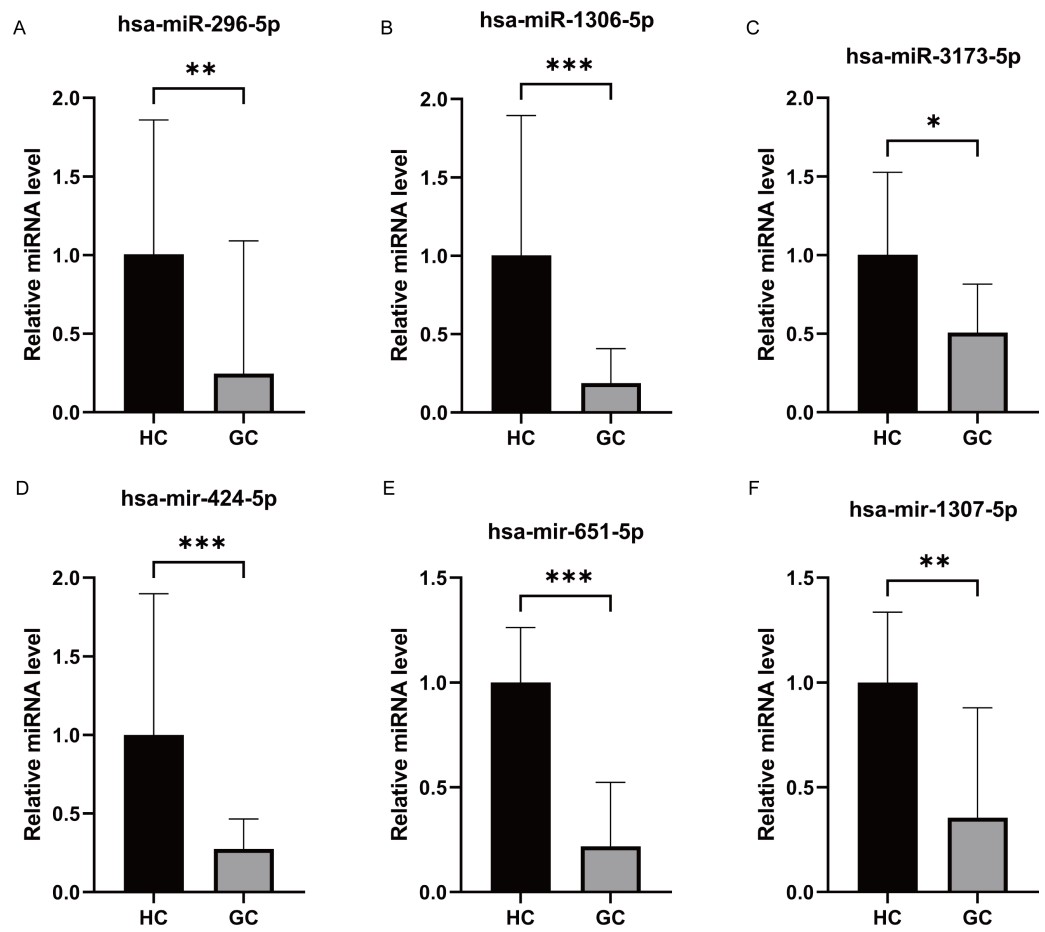

**Figure 2** **Differential expression of miRNAs analysis by RT-PCR.** (A–F) The qRT-PCR results revealed that the expression levels of hsa-miR-296-5p, hsa-miR-1306-5p, hsa-miR-3173-5p, hsa-miR-424-5p, hsa-miR-651-5p and hsa-miR-1307-5p. They all were significantly higher in the healthy group compared to the GC group. ***, $P < 0.001$; **, $P < 0.01$; *, $P < 0.05$.

as mean ± SEM). Based on the clinical staging of gastric cancer patients, they were categorized into two groups: stages 1–2 and stages 3–4. Subsequently, the ROC curves were generated for the three miRNAs based on these two groups, providing further evidence that hsa-miR-1306-5p, hsa-miR-3173-5p, and hsa-miR-296-5p serve as prognostic indicators for gastric cancer. The results demonstrate that the AUC values of hsa-miR-1306-5p, hsa-miR-3173-5p, and hsa-miR-296-5p are all above 0.7 (Table 2), indicating a relatively high predictive reliability. By determining the optimal cut-off values from the ROC curves of hsa-miR-1306-5p, hsa-miR-3173-5p, and hsa-miR-296-5p, we established that the cut-offs for has-mir-296-5p >0.160, has-mir-1306-5p >0.160, has-mir-3173-5p >0.930, and the combined predictor >0.622 ($p < 0.05$). Our findings highlight the close association of hsa-miR-1306-5p, hsa-miR-3173-5p, and hsa-miR-296-5p with the diagnosis and prognosis of gastric cancer, suggesting their potential as biomarkers for this disease.

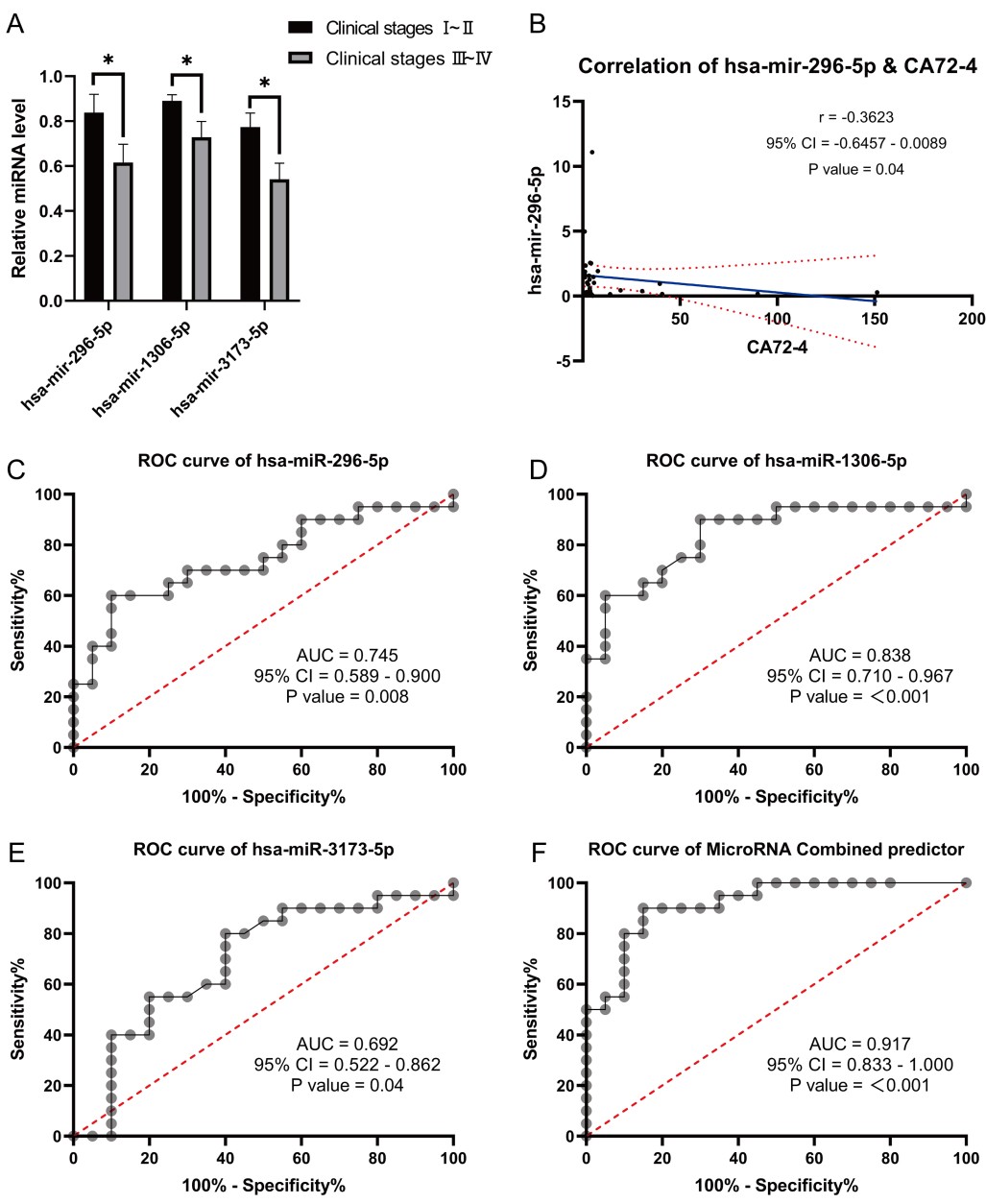

**Figure 3 Correlation analysis and Receiver Operating Characteristic (ROC) curves analysis.** (A) Correlation analysis between differentially expressed miRNA with clinical stages. (B) Correlation analysis of differentially expressed miRNA has-mir-296-5p with gastric cancer markers CA72-4. (C–F): Receiver Operating Characteristic (ROC) curves analysis were used to explore the possibility of these miRNAs expressing differentially in the blood as molecular markers. *, $P < 0.05$.

Correlation analysis between the expression of these three miRNAs and tumor markers CA724, CA199, and CEA revealed a negative correlation between hsa-miR-296-5p and CA724 ($p = 0.04$) (Fig. 3B). However, there were no significant correlations between hsa-miR-296-5p and CEA ($p = 0.32$), CA199 ($p = 0.49$); hsa-miR-1306-5p and CEA ($p = 0.23$),

**Table 2  Receiver Operating Characteristic (ROC) curves analysis of three miRNAs expressing differentially in the blood as molecular markers.**

| Predictor | AUC | *P* value | 95% CI | Cutoff value | Jordan index | Sensitvity |
|---|---|---|---|---|---|---|
| has-mir-296-5p | 0.841 | 0.01 | 0.623-1.000 | >0.160 | 0.714 | 1.000 |
| has-mir-1306-5p | 0.791 | 0.04 | 0.579-1.000 | >0.160 | 0.561 | 0.846 |
| has-mir-3173-5p | 0.802 | 0.03 | 0.606-0.997 | >0.930 | 0.539 | 0.539 |
| Combined predictor | 0.868 | 0.008 | 0.707-1.000 | >0.622 | 0.769 | 0.769 |

CA199 ($p = 0.37$), CA724 ($p = 0.10$); and hsa-miR-3173-5p and CEA ($p = 0.78$), CA199 ($p = 0.57$), CA724 ($p = 0.13$). Furthermore, we explored the possibility of using these differentially expressed miRNAs in the blood as molecular markers by performing ROC curve analysis. The results showed that the AUC values for hsa-miR-296-5p, hsa-miR-1306-5p, and hsa-miR-3173-5p in the blood of GC patients were 0.745, 0.838, and 0.693, respectively. The combined AUC of the three miRNAs was 0.917, demonstrating their potential effectiveness in predicting the occurrence of GC (Figs. 3C–3F).

## Differential expression of lymphocyte subsets in patients with GC

During the study period, a total of 51 GC patients were included in the experimental group. Among them, 30 were males and 21 were females, with an average age of ($69.02 \pm 1.36$) years. The healthy control group consisted of 50 individuals, with 29 males and 21 females, and an average age of ($60.50 \pm 1.49$) years. There was no statistically significant difference in age and gender distribution between the two groups ($P > 0.05$). Analysis of the expression changes in lymphocyte subsets between GC patients and the healthy control group revealed that the absolute numbers of CD3+ T cells, CD4+ T cells, CD8+ T cells, CD19+ B cells, and lymphocytes were lower in the GC group compared to the healthy control group (Figs. 4A–4F, expressed as mean ± SEM). However, the correlation analysis between the aforementioned differentially expressed lymphocyte subsets and clinical pathology did not yield significant differences ($P > 0.05$).

## ROC analysis results of blood lymphocyte subsets

Based on the significantly different lymphocyte subsets observed in GC patients and healthy individuals, ROC curves were plotted and the AUC values were determined. The ROC curve analysis revealed the following AUC values for GC patients: CD3+ T cells: 0.808 ($0.722-0.894$), CD4+ T cells: 0.776 ($0.681-0.871$), CD8+ T cells: 0.783 ($0.689-0.876$), CD19+ B cells: 0.836 ($0.748-0.923$), lymphocytes: 0.829 ($0.743-0.915$), and a combined ROC curve for all factors (CD3+ T, CD4+ T, CD8+ T, lymphocytes, and CD19+ B) with an AUC of 0.828 ($0.745-0.912$) (Figs. 5A–5F). Examining the correlation between the expression of the different factors (CD3+ T, CD4+ T, CD8+ T, lymphocytes, and CD19+ B) and tumor markers CA724, CA199, and CEA, it was found that CD4+T cells were negatively associated with CEA expression ($P = 0.03$). However, there was no significant correlation between CD4+T cells and CA199 ($P = 0.21$) or CA724 ($P = 0.11$). Similarly, no significant correlations were observed between CD3+ T cells and CEA ($P = 0.6$), CA199 ($P = 0.65$), or CA724 ($P = 0.37$); between CD8+ T cells and CEA ($P = 0.17$), CA199

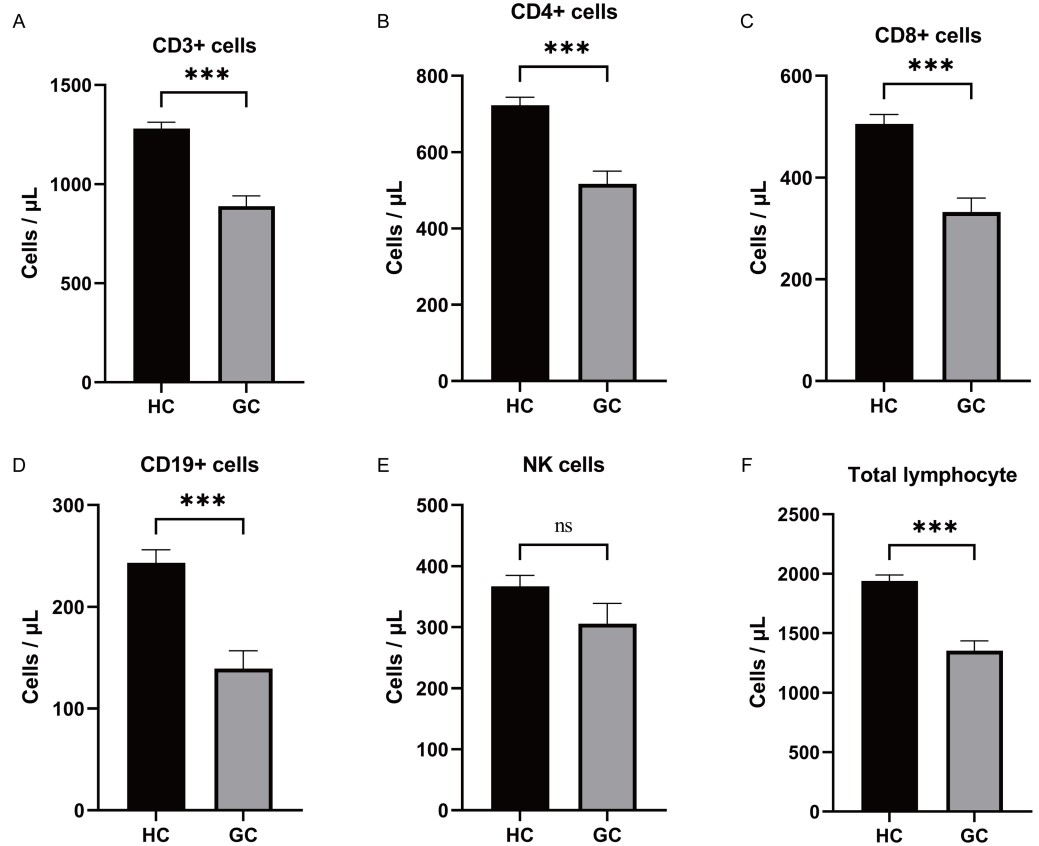

**Figure 4 Differential expression of lymphocyte subsets analysis.** (A–F) Analysis of the expression changes in lymphocyte subsets between GC patients and the healthy control group revealed that the absolute numbers of CD3+ T cells, CD4+ T cells, CD8+ T cells, CD19+ B cells, and total lymphocytes were lower in the GC group compared to the healthy control group. There was no significant difference in the expression of NK between the two groups. ***, $P < 0.001$; ns, not significant.

**Table 3 Logistic regression analysis of lymphocyte subsets (CD3+ cells, CD19+ cells) as independent predictors of gastric cancer.**

|  | $\beta$ | OR | 95% CI | $P$ value |
|---|---|---|---|---|
| CD3+ cells | −0.003 | 0.997 | 0.995–0.999 | 0.012 |
| CD19+ cells(log10 Transform) | −3.456 | 0.032 | 0.002–0.434 | 0.010 |
| Constant | 10.857 | 51,878.998 |  | <0.001 |

($P = 0.32$), or CA724 ($P = 0.84$); between CD19+ B cells and CEA ($P = 0.18$), CA199 ($P = 0.13$), or CA724 ($P = 0.86$); and between lymphocytes and CEA ($P = 0.56$), CA199 ($P = 0.77$), or CA724 ($P = 0.41$). Multivariate logistic regression analysis demonstrated that CD3+ T cells (odds ratio: 0.997 [0.995−0.999]) and CD19+ B cells (odds ratio: 0.032 [0.002−0.434]) can serve as independent predictors of GC (Table 3). Therefore, blood lymphocyte subsets can effectively predict gastric carcinogenesis and prognosis.

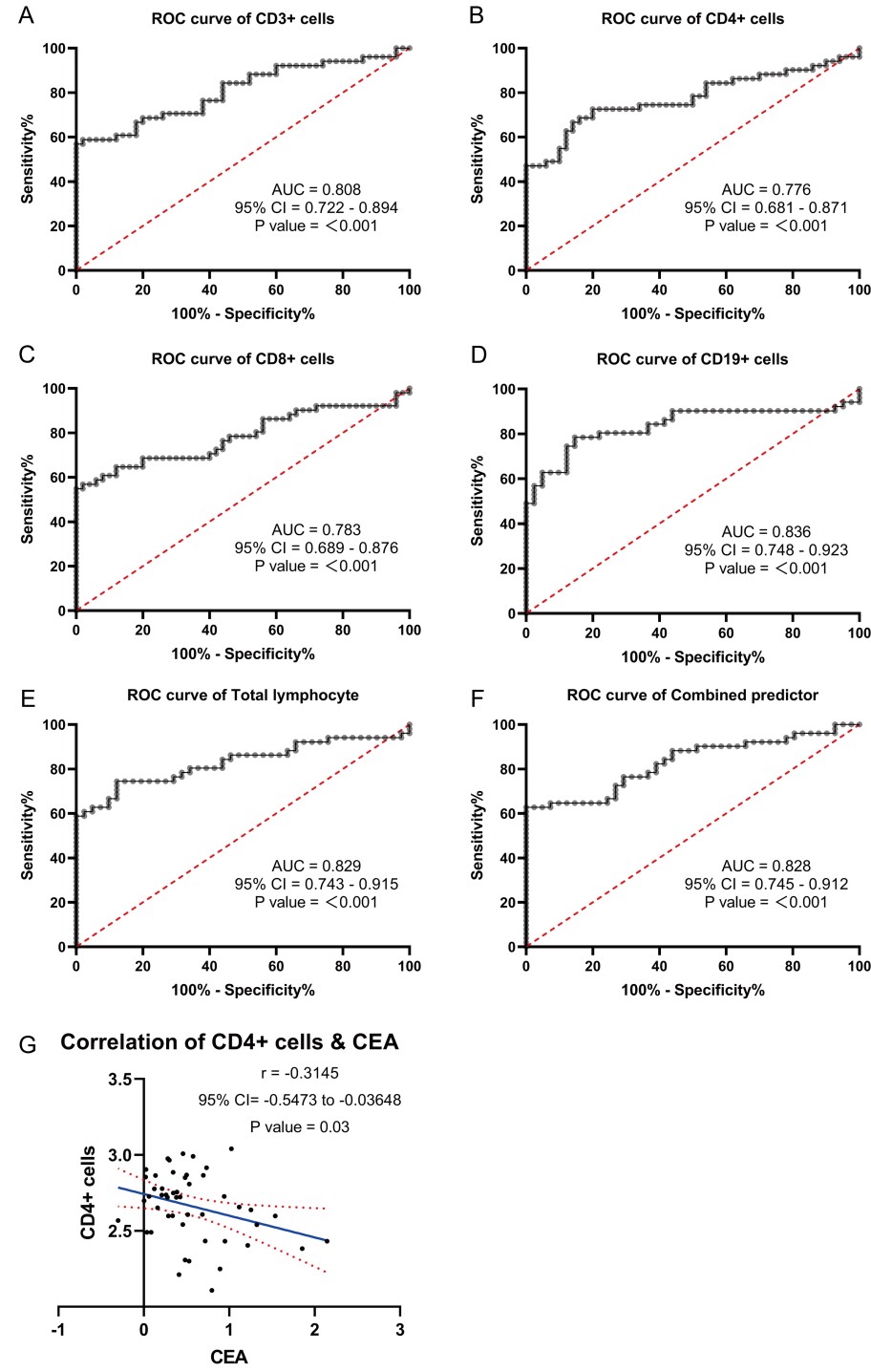

**Figure 5** ROC curve analysis were used to explore the possibility of these lymphocyte subsets express-ing differentially in the blood as molecular markers. (A) ROC curves of CD3+T. (B) ROC curves of CD4+T. (C) ROC curves of CD8+T. (D) ROC curves of CD19+B. (E) ROC curves of lymphocytes. (F) ROC curves of different factors. (G) Correlation analysis of differentially expressed lymphocyte subsets with gastric cancer markers CEA.

## DISCUSSION

GC is a significant global health issue and ranks as the third leading cause of cancer-related deaths (*Hohenberger & Gretschel, 2003*). However, current methods for detecting cancer in serum samples are limited. Although miRNAs were discovered by scientists over 20 years ago, they did not gain widespread attention until 2001 (*Kmiołek & Paradowska-Gorycka, 2022*). Since then, numerous human miRNAs have been identified, and their abnormal expression in tumor tissue has been linked to tumor growth, invasion, and metastasis (*Brennan & Henshall, 2020*; *Ferragut Cardoso et al., 2021*). Consequently, these aberrantly expressed miRNAs hold promise as potential early diagnostic markers for various stages of tumors (*Gomatou et al., 2021*; *Wang et al., 2021b*). Unfortunately, detecting miRNAs in tumor tissue involves invasive procedures and is not an optimal method for obtaining tumor samples (*Sengupta et al., 2021*). In 2008, a study suggested that miRNAs present in the blood of cancer patients could serve as biomarkers for predicting the occurrence of cancer (*Saliminejad et al., 2019*). Subsequently, miRNAs have been detected in other body fluids of cancer patients, offering a non-invasive approach for early tumor detection, including GC (*Orso et al., 2020*; *Vázquez-Mera et al., 2023*). *Tsujiura et al. (2010)* demonstrated that five miRNAs (miR-17-5p, miR-21, miR-106a, miR-106b, and let-7a) showed significant differences in serum samples of patients with preoperative GC compared to healthy controls. The diminished expression of KAI1 and heightened expression of miRNA-633 exhibit a substantial association with the unfavorable prognosis of malignant melanoma with gastric cancer, thereby establishing a foundation for considering KAI1 and miRNA-633 as potential novel molecular targets in the context of malignant melanoma with gastric cancer (*Wang et al., 2023*). The expression of miR-148a-3p was found to be significantly decreased in both the tissue and serum samples of patients with gastric cancer. Notably, serum miR-148a-3p demonstrated considerable potential as a diagnostic tool for gastric cancer. Furthermore, miR-148a-3p has been identified as an inhibitor of cancer progression and a novel biomarker for the diagnosis of gastric cancer (*Bao & Guo, 2020*). Shao's research showed that miR-212 expression is higher in healthy people than in those with gastric cancer (*Shao et al., 2020*). Wang's investigation underscored the promising prospects of miR-100, miR-125b, miR-199a, and miR-194 as prospective prognostic and diagnostic biomarkers within the realm of gastric cancer (GC) (*Wang et al., 2021a*). ROC analysis indicated that these miRNAs could serve as novel biomarkers for diagnosing GC. Furthermore, considering the heterogeneity of tumors, relying on a single miRNA for diagnosing GC in serum samples may not yield optimal results. Therefore, combining multiple miRNAs can significantly improve their diagnostic accuracy for GC. *Zhu et al. (2014a)* published a study on serum miRNA in 160 patients with GC and 160 healthy individuals. Their findings revealed that five miRNAs (miR-16, miR-25, miR-92a, miR-451, and miR-486-5p) were highly expressed in the blood of GC patients and showed potential as diagnostic biomarkers. Combining these five miRNAs resulted in higher accuracy for diagnosing early GC compared to using a single miRNA. *Zhu et al. (2019)* discovered that circNHSL1 functioned as a miR-1306-3p sponge, relieving miR-1306-3p's repressive impact on its target SIX1. In gastric cancer, miR-1306-3p was down-regulated and adversely linked

with pathological characteristics and poor prognosis. Six clinicopathologic variables and six microRNA expressions (miR-614, miR-1197, miR-4770, miR-3136, miR-3173, and miR-4636) were shown to be substantially different between TDs and non-TDs CRC patients in the SEER and TCGA training sets (*Xiao et al., 2022*). CircDIDO1 sponging miR-1307 reduced GC development by modulating the expression of the signal transducer inhibitor SOSC2. RGD-Exo-circDIDO1 could successfully transport circDIDO1 to GC cells, increasing SOCS2 expression (*Guo et al., 2022b*). It was discovered that the expression of miR-424-3p varied across healthy gastric, GC, and chemoresistant GC tissues. MiR-424-3p controlled the expression of ABCC2, a gene associated to chemoresistance. *Li et al. (2020)* found that miR-424-3p supported the development of GC's cancer and its resistance to chemotherapy. *Vaira et al. (2012)* discovered that miR-296 regulates a global gene signature for cell motility in epithelial cells by transcriptionally inhibiting the Scribble (Scrib) cell polarity-cell plasticity module. Loss of miR-296 leads to aberrantly elevated and mislocalized Scrib, which in turn promotes tumor cell invasiveness and random cell movement to be amplified. The expression of miR-651-5p in lung cancer tissues and cells was low. MiR-651-5p expression was correlated with tumor size, TNM stage, and lymph node metastasis but not with patient age or gender (*Lang et al., 2023*). Overall, the detection of serum miRNAs offers a non-invasive and promising approach for the diagnosis of GC (*Barry et al., 2018*). Multiple miRNAs in combination hold great potential to enhance the diagnostic capabilities for early detection of GC (*Zhu et al., 2014b*).

In this study, we initially screened 50 differentially expressed miRNAs between healthy individuals and patients with GC using second-generation miRNA sequencing. Subsequently, we performed bioinformatics analysis to identify 6 miRNAs strongly correlated with GC. We then validated the expression of these miRNAs in the blood of GC patients using qRT-PCR. Among them, hsa-miR-1306-5p, hsa-miR-3173-5p, and hsa-miR-296-5p were significantly downregulated in the GC group compared to the healthy group. Conversely, the signal values of hsa-miR-424-5p, hsa-miR-651-5p, and hsa-miR-1307-5p were significantly higher in the GC group. These qRT-PCR results confirmed the downregulation of hsa-miR-1306-5p, hsa-miR-3173-5p, and hsa-miR-296-5p in the blood of GC patients, consistent with the miRNA sequencing data. It is worth noting that the sample size of GC blood specimens included in this study may have certain limitations, and further validation with an expanded sample set is warranted. Furthermore, the analysis of miRNA expression and clinical characteristics of GC patients revealed that the expression levels of hsa-miR-1306-5p, hsa-miR-3173-5p, and hsa-miR-296-5p were correlated with the stage of GC. Specifically, as the cancer stage advanced, the expression of these miRNAs decreased. Additionally, when examining the correlation between the expression of these three miRNAs and tumor markers CA724, CA199, and CEA, a negatively correlation was observed between hsa-miR-296-5p and CA724. To explore the potential of these miRNAs as molecular markers for predicting the occurrence of GC, we performed ROC curve analysis. Interestingly, the AUC values for hsa-miR-1306-5p, hsa-miR-3173-5p, and hsa-miR-296-5p in the blood of GC patients were 0.838, 0.693, and 0.745, respectively. Moreover, the combined AUC of the three miRNAs was 0.918, indicating their effectiveness in predicting the occurrence of GC.

The findings of this study highlight the potential significance of hsa-miR-1306-5p, hsa-miR-3173-5p, and hsa-miR-296-5p in the early diagnosis, prognosis, and monitoring of GC. Tumor development is closely associated with disturbances in the body's immune balance, leading to alterations in the number and proportion of immune cells. Lymphocytes and their subsets play a crucial role in maintaining immune system functionality. Similar to immune diseases and other infectious conditions, cancer can disrupt the balance of lymphocyte subsets (*Wang et al., 2020*). Cell surface molecules, such as CD3+, CD4+, CD8+, NK, and CD19+, are involved in humoral immunity and cytotoxic immune responses against tumors. Therefore, investigating the characteristics of lymphocyte subsets in cancer patients can provide valuable insights into the mechanisms by which cancer affects the body's immune system. *Chen et al. (2021)* reported that in patients with advanced esophageal cancer, the proportion of T lymphocytes, CD4+ T lymphocytes, CD8+ T lymphocytes, and natural killer cells in the draining lymph nodes was significantly reduced compared to early-stage patients. Conversely, the proportion of B cells and T-reg cells was significantly increased. However, the ratio of CD4+/CD8+ T cells did not show any statistical significance. These findings suggested a disruption of lymphocyte subsets at the tumor site, which was associated with lymph node metastasis. In another study, *Jiang et al. (2019)* observed a significant increase in the proportion of CD4+ memory T cells, CD8+ memory T cells, and Treg cells in the peripheral blood of GC patients. These cell types were also correlated with lymph node metastasis. The authors concluded that patients with GC exhibited an immunosuppressive state in the presence of tumors. In our study, we found that the absolute numbers of lymphocytes, CD3+ T cells, CD4+ T cells, CD8+ T cells, and CD19+ B cells were lower in the GC group compared to the healthy control group ($P < 0.05$). However, there was no significant difference in the CD4+/CD8+ ratio ($P > 0.05$). Additionally, we investigated the correlation between the expression of these different factors (lymphocytes, CD3+ T, CD4+ T, CD8+ T, and CD19+ B) and tumor indicators CA724, CA199, and CEA. We found that CD4+ T cells were associated with CEA expression, indicating that older patients with lower tumor differentiation tended to have higher CEA expression and worse overall survival rates.

Furthermore, ROC curve analysis revealed the AUC values for lymphocytes, CD3+ T cells, CD8+ T cells, CD4+ T cells, and CD19+ B cells in GC patients to be 0.829 (0.743−0.915), 0.808 (0.722−0.894), 0.783 (0.689−0.876), 0.776 (0.681−0.871), and 0.836 (0.748−0.923), respectively. The combined AUC of these factors was 0.828 (0.745−0.912). Moreover, multivariate logistic regression analysis demonstrated that CD3+ T cells (odds ratio: 0.997 [0.995−0.999]) and CD19+ B cells (odds ratio: 0.032 [0.002−0.434]) could be used as independent predictors of GC.

Despite the promising nature of our findings, it is important to acknowledge the presence of various limitations within our study: (1) Given the limited sample size, it is imperative to conduct additional large cohort validations. (2) Normalization plays a crucial role in achieving precise relative quantification of miRNA levels using qRT-PCR. However, there is currently no consensus on universally accepted internal controls for measuring serum miRNA. Therefore, employing an absolute quantification approach would be more advantageous for subsequent validation endeavors; and (3) The authors concluded that the

number of patients in this study was limited, and that larger clinical trials and/or further studies with large sample sizes are needed next to further validate the value and significance of miRNAs as biomarkers for the diagnosis of gastric cancer, as well as to carry out deeper mechanistic studies.

## CONCLUSIONS

Currently, the detection of miRNA and T lymphocyte subsets in blood is not routinely performed in clinical practice for tumor patients, and there is a lack of prospective studies investigating the specific changes and impact of immune indicators on tumor occurrence and prognosis. In summary, we identified that certain abnormal proportions of miRNAs (hsa-miR-1306-5p, hsa-miR-3173-5p, hsa-miR-296-5p) and lymphocyte subsets (the absolute numbers of CD3+ T cells, CD4+ T cells, CD8+ T cells, CD19+ B cells, total lymphocytes) are strongly associated with the diagnosis and prognosis of GC, making them potential biomarkers for this disease. Consequently, monitoring the alterations in miRNA and T lymphocyte subsets in blood could aid clinicians in evaluating the progression and prognosis of GC patients, and it holds promise for future clinical applications.

## ACKNOWLEDGEMENTS

The authors wish to thank the staff of department of Laboratory, Respiratory Medicine, Infection Department, Putuo Hospital, Shanghai University of Traditional Chinese Medicine for their participation.

### Funding
This work was supported by the Shanghai Science and Technology Plan Project General Project 23ZR1456400 (Zixi Chen), the Shanghai Municipal Commission of Health and Health Clinical Subject Youth Project 20214Y0502 (Zixi Chen), and the National Natural Science Foundation of China 31800988 (Zixi Chen). The funders had no role in study design, data collection and analysis, decision to publish, or preparation of the manuscript.

### Grant Disclosures
The following grant information was disclosed by the authors:
Shanghai Science and Technology Plan Project General Project: 23ZR1456400.
Shanghai Municipal Commission of Health and Health Clinical Subject Youth Project: 20214Y0502.
National Natural Science Foundation of China: 31800988.

### Competing Interests
The authors declare there are no competing interests.

## Author Contributions

- Jinpeng Li conceived and designed the experiments, prepared figures and/or tables, and approved the final draft.
- Zixi Chen conceived and designed the experiments, prepared figures and/or tables, and approved the final draft.
- Qian Li conceived and designed the experiments, performed the experiments, prepared figures and/or tables, and approved the final draft.
- Rongrong Liu conceived and designed the experiments, performed the experiments, prepared figures and/or tables, and approved the final draft.
- Jin Zheng conceived and designed the experiments, analyzed the data, prepared figures and/or tables, and approved the final draft.
- Qing Gu conceived and designed the experiments, analyzed the data, prepared figures and/or tables, and approved the final draft.
- Fenfen Xiang conceived and designed the experiments, prepared figures and/or tables, and approved the final draft.
- Xiaoxiao Li conceived and designed the experiments, prepared figures and/or tables, provision of reagents and laboratory assistance, and approved the final draft.
- Mengzhe Zhang conceived and designed the experiments, prepared figures and/or tables, provision of reagents and laboratory assistance, and approved the final draft.
- Xiangdong Kang conceived and designed the experiments, prepared figures and/or tables, authored or reviewed drafts of the article, and approved the final draft.
- Rong Wu conceived and designed the experiments, prepared figures and/or tables, authored or reviewed drafts of the article, and approved the final draft.

## Ethics

The following information was supplied relating to ethical approvals (i.e., approving body and any reference numbers):

The study was approved by the Ethics Administration Office of Pu Tuo Hospital, Shanghai University of Traditional Chinese Medicine (No. PTEC-A-2022-31-1).

## Field Study Permissions

The following information was supplied relating to field study approvals (i.e., approving body and any reference numbers):

The study was approved by the Ethics Administration Office of Pu Tuo Hospital, Shanghai University of Traditional Chinese Medicine (No. PTEC-A-2022-31-1).

## DNA Deposition

The following information was supplied regarding the deposition of DNA sequences:

The sequences are available at GenBank: PRJNA988889.

## Data Availability

The raw data are available in the Supplemental Files.

## Supplemental Information

Supplemental information for this article can be found online at http://dx.doi.org/10.7717/peerj.16660#supplemental-information.

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
