# Peer review of "Study of miRNA and lymphocyte subsets as potential biomarkers for the diagnosis and prognosis of gastric cancer"

_PeerJ, doi:10.7717/peerj.16660_

## Round 0.1 · original submission · Major Revisions

The most important issues are the small sample size and the missing clinical criteria used for prognostic. All the suggestions from reviewers have to be addressed to revise the manuscript.

Reviewer 1 ·

Basic reporting

The manuscript titled “Study of miRNA and lymphocyte subsets as potential biomarkers for the diagnosis and prognosis of gastric cancer” was done by Li et al., it need some revisions.

Major revision
1.Conclusion In abstract, “These miRNAs and lymphocyte subsets”, the author should give the accurate information such as hsa-miR-1306-5p and CD8+ T cells et al.
2.The conclusion in line 355-362 should be rewritten.
3.In my opinion, line 349-352 can be deleted.

Minor revision
1.Line 108, please to add a space between (3) and All selected in “(3)All selected--”
2.Line 110, “Organization Helsinki Declaration..”, deleted a period.

Experimental design

no comment

Validity of the findings

no comment

Additional comments

no comment

Reviewer 2 ·

Basic reporting

The manuscript is well written. Sufficient background information is provided. However, the details provided are very superficial. Specifically, if authors are aiming to decipher novel biomarkers, an in-depth understanding of the disease, technique, state-of-the-art, limitations thereof, and gaps in the knowledge need to be identified.
The authors fail to mention several studies that have identified miRNAs that have shown statistically significant differences in the plasma of gastric cancer patients compared to healthy individuals in the background information. The authors mention a couple of these studies in the discussion section but their coverage of the relevant literature is minimal.

Experimental design

Although the research question is of high importance, what contributions the authors aim to make through their work is not clear. Several studies have been published that have identified miRNAs differentially found in the plasma samples of gastric cancer patients compared to healthy individuals - many of these studies have large sample sizes, better positive and negative control, and robust statistical analyzes than what the current study offers.

The sample size is way too small to draw any meaningful conclusions from it.

Samples are not well controlled.

Samples are not well defined - for example, what were the clinically defined stages for these gastric cancer patients? Were there other medical conditions along with GC?

The authors indicate that most of the GC patients underwent some kind of medical intervention and these are significant interventions known to affect whole-body physiological and biochemical parameters. Under these conditions, using these samples for immune profiling and further analysis is not scientifically valid.

Considering it would be relatively easy to find healthy controls, why did the authors not use similar gender profiles in healthy individuals compared to GC samples?

The samples collected are way too narrow in geographical location to use these samples and make bold claims such as the identification of biomarkers that can correlate to stages of GC.

For something to be used as a biomarker, a lot more controls are needed. For example, did authors use patient samples that have been diagnosed with some other type of cancer and underwent similar medical interventions as GC patients in this study? This would be necessary to ensure these biomarkers' are GC-specific.

qPCR experiments were not well controlled. Resultant data is not very useful to draw any meaningful conclusions.

Sample sizes are different and not well-defined for lymphocyte analyses.

These authors mention several sets of miRNAs proposed as biomarkers for GC. What were the limitations of those studies?

What are the limitations of the current set of biomarkers for GC?

Validity of the findings

Due to the multiple limitations of the experimental design and study design identified above, the observations and conclusions are not scientifically robust. The sample size is way too small. Statistical analyses are not robust.

Reviewer 3 ·

Basic reporting

- The authors stated that hsa-miR-296-5p was positively correlated with
CA724. However, in the abstract-results, they also mentioned has-miR296-5p was found to be negatively correlated with CA724.

Experimental design

- What criteria did the authors use for prognostic model?
- Line 109, to the ethical principles outlined in the World Health Organization
Helsinki Declaration.. it is not correct, it should be written: … outlined in the
Declaration of Helsinki.

Validity of the findings

In the discussion section, the authors did not compare their findings with
the other works sufficiently, especially with the findings of miRNAs, hsamiR-1306-5p, hsa-miR-1307-5p, hsa-miR- 3173-5p, hsa-miR-424-5p,
hsa-miR-651-5p and hsa-miR-296-5p.

Additional comments

If possible, please add limitations of this assay

---

## Round 0.2 · Minor Revisions

The small number of samples is the major limitation of this study. But, GC patients considered in this study didn't receive therapies and this is quite an important condition (supporting the small amount of samples). Either from Reviewer and authors an important confusion emerged: the study has been performed in serum. Please check carefully this aspect in the materials and methods section. Especially related to qRT-PCR, the samples used cannot be in Eparin (EP tubes!!!) because it interferes with the analysis.

I also suggest changing the title from "Verification of miRNAs by qRT-PCR" to "Validation of miRNAs by qRT-PCR". Furthermore, in the same section, please specify how the data have been represented, I guess it is through 2-DCt method.

---

## Round 0.3 · Minor Revisions

Please, make a final check on the 2-ΔΔCt method used. This represents fold change related to the control considered. This latter should always have the value of 1. This is why I think that probably the 2-ΔCt method has been used (fold change related to U6).

---

## Round 0.4 · accepted · Accept

The manuscript has been revised properly and is ready for publication.

Reviewer 1 ·

Basic reporting

The manuscript titled “Study of miRNA and lymphocyte subsets as potential biomarkers for the diagnosis and prognosis of gastric cancer”was done by Jinpeng Li et al.The author clarified all my concerns.It can be accepted for publishing.

Experimental design

no

Validity of the findings

no

Additional comments

no